# Studies on the Volumetric Stability and Mechanical Properties of Cement-Fly-Ash-Stabilized Steel Slag

**DOI:** 10.3390/ma14030495

**Published:** 2021-01-21

**Authors:** Mingkai Zhou, Xu Cheng, Xiao Chen

**Affiliations:** 1State Key Laboratory Silicate Materials for Architecture, Wuhan University of Technology, Wuhan 430070, China; silicate@whut.edu.cn; 2School of Materials Science and Engineering, Wuhan University of Technology, Wuhan 430070, China; 15172488673@163.com

**Keywords:** road base, steel slag, expansion

## Abstract

The stability of steel-slag road materials remains a critical issue in their utilization as an aggregate base course. In this pursuit, the present study was envisaged to investigate the effects of fly ash on the mechanical properties and expansion behavior of cement-fly-ash-stabilized steel slag. Strength tests and expansion tests of the cement-fly-ash-stabilized steel slag with varying additions of fly ash were carried out. The results indicate that the cement-fly-ash-stabilized steel slag exhibited good mechanical properties. The expansion rate and the number of bulges of the stabilized material reduced with an increase in the addition. When the addition of fly ash was 30–60%, the stabilized material was not damaged due to expansion. Furthermore, the results of X-CT, XRD and SEM-EDS show that fly ash reacted with the expansive component of the steel slag. In addition, the macro structure of the stabilized material was found to be changed by an increase in the concentration of the fly ash, in order to improve the volumetric stability. Our study shows that the cement-fly-ash-stabilized steel slag exhibits good mechanical properties and volumetric stability with reasonable additions of fly ash.

## 1. Introduction

In recent years, there has been a huge demand for sand and gravel aggregates due to the rapid development of highway construction in China and the exhaustion of the natural aggregate resources. Therefore, it is necessary to find new materials that are economically and environmentally friendly, and that can replace the natural aggregate. In addition, as a by-product of the steel-making process, the annual emission of steel slag in China exceeds 100 million tons, but the utilization rate is only 20%, which is far lower than that of Europe and America [1,2].

Steel slag is an active aggregate with low crushing value and good wear resistance, and its main application is its use as an aggregate base course [3,4]. At present, there are many studies on and applications of steel-slag base-course materials. Liu [5] reported that using steel slag instead of crushed stone can improve the mechanical properties of the cement-stabilized macadam. Li [6] compared the dry-shrinkage and temperature-shrinkage characteristics of cement-stabilized steel slag and cement-stabilized macadam, and found that the cement-stabilized steel slag with rational adjustment of gradation exhibited a smaller dry-shrinkage strain and did not produce a lager temperature-shrinkage strain. Even though the performance of the cement-stabilized steel slag material is excellent and meets the Chinese criteria, the practical application effect of steel-slag base materials is not feasible. In the early stage, the steel-slag base course built by Wuhan Iron and Steel Group and Baosteel Group in China exhibited severe fractures after six months [7]. Steel slag from a steel plant in Liaoning Province, China was used for the construction of the cement-stabilized steel slag base course. This section of road was damaged by expanding after 90 days, even though the steel slag was qualified [8]. These problems have severely restricted the use of steel slag in base courses.

The instability of steel slag results from the fact that it contains free calcium oxide and free magnesium oxide, which are the direct cause of destruction in steel slag base courses. The volume of Ca(OH)_2_ and Mg(OH)_2_ generated by the hydration of free calcium oxide and free magnesium oxide increased by 91.7% and 119.6%, respectively, which caused the stress concentration of the cement-base materials and produced microcracks [9,10]. At present, steel slag needs to be subjected to a water-immersion expansion test before its engineering application to assess its volumetric stability (requiring a water-immersion expansion rate <2%). Even if the steel slag meets the requirements, the expansion of some steel slag particles enriched with f-CaO can easily cause local damage to the steel-slag base due to poor homogeneity.

The pretreatment of steel slag can alleviate its instability, but there are technical and economic barriers in the natural aging treatment [11,12], autoclaved treatment [13], carbonization treatment [14,15], and high-temperature modification [16] that are difficult to apply on a large-scale steel-slag treatment. Some studies have shown that active SiO_2_ and Al_2_O_3_ of mineral admixture can react with the hydration products of the f-CaO of steel slag [17,18,19]. On one hand, the activity of the mineral admixture can be effectively excited to enhance the strength of the material; on the other hand, the expansion caused by f-CaO of steel slag can be alleviated. Currently, there are some studies on fly-ash-stabilized steel-slag base courses. Sharma [20] and Shen [21] found that the strength development of the fly-ash-stabilized steel-slag material is the same as that of lime-fly-ash-stabilized steel-slag material, which has a good long-term strength. The incorporation of phosphogypsum or lime can promote a substantial increase in strength. However, the early strength of fly-ash-stabilized steel-slag base material is low, and there are still many difficulties in the use of phosphogypsum and lime, such as the pretreatment process and batching. Xu Fang [22] indicated that the mechanical indices of cement-fly-ash-stabilized steel slag are more prominent than those of lime-fly-ash-stabilized steel slag and cement-stabilized steel slag. In general, the research on fly-ash-stabilized steel-slag material has mainly focused on the composition design and mechanical properties of materials, the expansion caused by the heterogeneity of steel slag, and its influence on the volumetric stability of steel-slag base-course material need to be explored.

Therefore, the present study focused on the influence of the addition of fly ash on the expansion behavior and mechanical properties of cement-stabilized steel slag. Using a high-definition three-dimensional X-ray microscope (X-CT), a scanning electron microscope (SEM), and energy-dispersive spectroscopy (EDS), the inhibition principle of fly ash on the expansion behavior of steel slag was revealed from the macro structure to the micro product formation.

## 2. Materials and Methods

### 2.1. Materials

Steel slag was supplied from Taiyuan Iron and Steel Group (Taiyuan, China). The chemical composition of steel slag was detected by X-ray fluorescence ( PANalytical.B.V, Almelo, The Netherlands), as shown in Table 1. The X-ray diffraction analysis of the steel slag was conducted as shown in Figure 1. The refinement of the particle size in steel slag can improve its uniformity [23], and considering the economic factors and homogeneity comprehensively, the particle size was in the range of 0–10 mm. The particle-size distribution is shown in Figure 2. The basic physical and chemical properties of steel slag are shown in Table 2. The water immersion expansion rate was 0.108%, which was far lower than the limit value (test method GB/T 24175-2009 of China, which is very similar to ASTM D4792; required water-immersion expansion rate <2%).

The fly ash was obtained from the Thermal Power Plant in Shanxi Province, China. Its chemical composition is shown in Table 1. The cement was a PO42.5 ordinary Portland cement produced by Shanxi Cement Company (China). Its chemical composition is shown in Table 1, and its physical and mechanical properties are shown in Table 3.

### 2.2. Mix Formulas and Experimental Method

Seven kinds of cement-fly-ash-stabilized steel slag with different fly ash additions were designed, and their unconfined compressive strength, splitting strength, compressive resilience modulus, and material-expansion rate were tested, in order to explore the influence of fly ash on the expansion behavior and mechanical properties of cement-stabilized steel slag. Moreover, the macro structure of the material was observed by using the X-CT (Zeiss, Jena, Germany), the reaction products of the steel slag and cement-fly-ash binder were studied using the SEM-EDS (Zeiss, Jena, Germany), and its mineral composition was detected by XRD (Bruker AXS, Leipzig, Germany). The expansive product of the steel slag was also detected by XRD.

#### 2.2.1. Mix Formulas

The mix proportion of the cement-fly-ash-stabilized steel slag is listed in Table 4. The cement content was 4%, and the steel slag was replaced by fly ash of the same quality. The additions of fly ash were 0, 10%, 20%, 30%, 40%, 50%, 60%, numbered as 0FASS, 10FASS, 20FASS, 30FASS, 40FASS, 50FASS, and 60FASS, respectively. According to the Chinese standard (JTG E51-2009), the maximum dry density and optimum moisture content of the mixture were determined by the compaction test. In addition, referring to the method of Chen [24], the ratio of the volume of cement-fly-ash binder to the void volume of compacted steel slag aggregate was defined as filling factor λ. When λ < 1, the binder did not completely fill the aggregate voids, and the stabilized material was a skeleton-pore type; when λ ≈ 1, the binder was filled with aggregate voids, and the stabilized material was a skeleton-dense type; When λ > 1, the skeleton structure was destroyed, and the stabilized material was a suspended-dense type. The filling factor λ of 0FASS, 10FASS, 20FASS, 30FASS, 40FASS, 50FASS, and 60FASS were calculated by this method, and the corresponding structure types are listed in Table 4.

#### 2.2.2. Strength Test

The well-mixed samples were compressed into a Φ100 mm × 100 mm cylinder with a compaction degree of 98%. After demolding, the samples were cured in a room at a temperature of 20 ℃ and a relative humidity of 98%. Before the strength test, the sample was soaked in water for 24 h before curing to the test age. According to the Chinese standard (JTG E51-2009), the unconfined compressive strength, splitting strength, and compressive modulus of resilience of the specimens were tested. The test ages were 7 days, 28 days, and 90 days.

#### 2.2.3. Expansion Behavior of Materials under Standard Curing Conditions

Although the test results of the water-immersion expansion rate of the steel slag were qualified before the application of the steel slag, research has shown that steel-slag base material prepared with this qualified steel slag still exhibits some defects, such as bulging and cracking [25]. Therefore, we observed the expansion behavior of the cement-fly-ash-stabilized steel slag in the standard curing condition, in order to better simulate its expansion behavior in the actual service process. According to the test method of the Chinese standard (JTG E51-2009), the size of the specimen was 100 mm × 100 mm × 400 mm with a compaction degree of 98% (Figure 3). One side of the sample was close to the left baffle of the instrument, and the other side of the sample was bonded with a glass sheet. The dial indicator head contacted with the glass piece to detect the change in sample length. The sample was tested in a room with a temperature of 20 ℃ and a relative humidity of 98%, and the dial gauge readings were recorded for 180 days. In addition, the number and the description of bulges of the materials were recorded. Cement-fly-ash-stabilized steel slag with fly ash additions of 0, 10%, 20%, 30%, 40%, 50% and 60% (0FASS, 10FASS, 20FASS, 30FASS, 40FASS, 50FASS, and 60FASS, respectively) were tested. There were three parallel samples in each group. The average value of the expansion rate was used to characterize the degree of material volumetric change, and the dispersion coefficient was used to characterize the uniformity of volumetric change rate of three parallel samples.

#### 2.2.4. X-CT

A high resolution three-dimensional X-ray microscope (Xradia 510 Versa) was used to obtain the images of the macro structure of the cement-fly-ash-stabilized steel-slag material. According to the Chinese standard (JTG E51-2009), the sample size was Φ 50 mm × 50 mm. The cube range of 30 mm × 30 mm × 30 mm inside the sample was scanned by employing a high-resolution three-dimensional X-ray microscope. The research object was the cement-fly-ash-stabilized steel slag with fly ash additions of 0, 20%, and 40% (0FASS, 20FASS, and 40FASS, respectively).

#### 2.2.5. XRD and SEM-EDS

In order to explore the principle of the interaction between the steel slag and the cement-fly ash binder, after the expansion test of the cement-fly-ash-stabilized steel slag material with 20% fly ash content (20FASS), the steel-slag particles were stripped at the place where the bulge occurred. There were cracks in the contact between the steel slag and another aggregate particle. The surface of the steel slag and the cracks were covered by white powder, and the other side of the steel-slag particle was wrapped by the cement-fly-ash binder. The composition of the white powder was detected by XRD. The morphology and element distribution of the cement-fly-ash binder on the side, where the binder contacted the steel slag, were observed by SEM-EDS, and its mineral composition was detected by XRD, so as to reveal the chemical interaction between the steel slag and cement-fly-ash binder. The flow chart of the Sample handling and testing process are shown in Figure 4.

## 3. Results and Discussion

### 3.1. Effect of Fly Ash on the Mechanical Properties of Cement-Stabilized Steel Slag

Figure 5, Figure 6 and Figure 7 show the unconfined compressive strength, splitting strength, and compressive modulus of resilience of the cement-fly-ash-stabilized steel slag with various additions of fly ash (0, 10%, 20%, 30%, 40%, 50%, and 60%).

Figure 5 shows that with an increase in fly ash, the compressive strength of the cement-fly-ash-stabilized steel-slag material first increased, and later decreased. It reached a maximum value when the fly-ash addition was 20% (20FASS). Compared with the cement-stabilized steel slag (0FASS), the 7-day, 28-day, and 90-day unconfined compressive strength of 20FASS was found to be increased by 126%, 157%, and 197%, respectively. This was due the binder coating on the surface of steel-slag particles being relatively thin without adding fly ash, resulting in a small contact area between the steel-slag particles, which easily caused stress concentration and damage under pressure. With the addition of fly ash, the thickness of the binder coating gradually increased. Due to the use of the compaction molding, fly ash exhibited a good fixation effect around the contact area between the steel-slag particles to disperse and bear the compressive stress. However, with the continuous increase of the fly-ash content, the structure changed from skeleton-type to suspension-type, which can not only destroy the interlocking effect of steel slag, but also reduces the strength of the binder due to a decrease in the cement proportion, and eventually leads to a decrease in the compressive strength of the materials.

Figure 6 shows that the variation of the splitting strength with the fly-ash content was the same as that of the unconfined compressive strength, and the splitting strength also reached a peak at 20% fly-ash addition (20FASS) and subsequently decreased. Figure 7 shows that the 7-day and 28-day resilient modulus of 10FASS were higher than that for 20FASS, indicating that the skeleton effect had a great influence on the resilient modulus. With an increase in the fly-ash content, the 7-day and 28-day resilient modulus gradually decreased. This was due to the fact that the increase in the fly-ash addition weakened the skeleton effect of the stabilized material and the strength of the binder.

### 3.2. Effect of Fly Ash on Expansion Behavior of Cement-Stabilized Steel Slag

Figure 8a–g shows the expansion rates of 10FASS, 20 FASS, 30 FASS, 40 FASS, 50 FASS, and 60 FASS, respectively. Table 5 shows the expansion rate, average value of expansion rate, dispersion coefficient of expansion rate, and the number and the description of bulges of the stabilized materials when the age was 180 days.

Figure 8a shows the expansion rate of the cement-stabilized steel slag (0FASS). From the results, it can be seen that the increment of volumetric change can be divided into three stages: high speed, medium speed, and stable. The high-speed stage, from 0–20 days, was due to the rebound of materials after compression and the hydration of free calcium oxide in steel slag. In the medium-speed stage, from 20–100 days, the amount of free calcium oxide gradually decreased with the hydration process, which decreased the formation rate of the expansive products. The stable stage, from 100–180 days, showed almost no change in the volume of the material.

Figure 8 and Table 5 show that the average value of the expansion rate decreased with an increase in the fly-ash content. This is because the steel slag was replaced by the fly ash, which reduced the proportion of steel slag in the stabilized material. The dispersion coefficient decreased with the increase of fly-ash content, and the dispersion coefficient of the skeleton type (0FASS, 10FASS, and 20FASS) was much higher than that of the suspension type (30FASS, 40FASS, 50FASS, and 60FASS). This was related to the bulges on the surface of some samples of 0FASS, 10FASS, and 20FASS. The bulges directly pushed on the glass plate, which led to a sharp increase in the expansion rate of these samples. A similar situation occurred in Arlyn’s [26] research on the expansion law of steel-slag-stabilized soil. On the contrary, the suspended cement-fly-ash-stabilized steel slag exhibited a smooth surface without bulge.

In general, Figure 8 and Table 5 suggest that an increase in fly ash enhances the inhibition on expansion and bulge of the stabilized material. It is related to the structure change of the stabilized materials and the reaction between the fly ash and steel slag. In addition, on the 40th day of the experiment, 40FASS, 50FASS, and 60FASS exhibited a volumetric shrinkage, which was related to the autogenous shrinkage of a large amount of fly-ash material [27].

### 3.3. Inhibition Mechanism of Fly Ash on Expansion Behavior of Cement-Stabilized Steel Slag

#### 3.3.1. Inhibition Mechanism of Macro Structure on Expansion Behavior of Cement-Fly-Ash-Stabilized Steel Slag

Figure 9a–f shows the X-CT images of 0FASS, 20FASS, and 40FASS; Figure 10a shows the simulation diagram of the cement-fly-ash-stabilized steel slag material (skeleton type); and Figure 10b presents the simulation diagram of the cement-fly-ash-stabilized steel-slag material (suspension type).

Figure 9a–d show that the steel-slag particles in the stabilized material (0FASS and 20FASS) contacted each other, and the steel-slag particles formed multiple skeletons (blue represents the cement-fly-ash binder and pore with a lower relative density, and other granular colors indicate steel-slag particles with a higher relative density). Steel slag has a high rigidity, and its elastic modulus was more than 80 GPa [28,29]. Figure 10a shows that the expansion stress generated by the expansion of the steel slag in the skeleton, which directly pushed the steel-slag particles, leading to bulging and cracking. As shown in Figure 9e,f, the steel-slag particles of the 40FASS were dispersed in the cement-fly-ash binder to form a suspension type. There was almost no direct contact between the steel-slag particles, which reduced the probability of the steel slag forming a skeleton in the stabilized material.

Moreover, the materials with a low elastic modulus are known to produce an elastic-plastic deformation to accommodate the expansive components of other materials [30,31,32,33]. The 28-day elastic modulus of the cement-fly-ash binder with high content of fly ash was found to be less than 1 GPa [34,35], while the 28-day elastic modulus of the cement binder was more than 20 GPa [36]. Compared with the cement binder alone, the cement binder with a large amount of fly ash exhibited characteristics of low elastic modulus, low brittleness and high creep compliance at early stages [37,38,39,40]. Figure 10b shows that the steel-slag particles of the stabilized material (suspension type) were wrapped by the binder. For steel-slag particles enriched with f-CaO, the cement-fly-ash binder with high addition of fly ash produced an elastic-plastic deformation to accommodate the expansive component of the steel slag, and slowed the expansion stress.

#### 3.3.2. Reaction Mechanism of Fly Ash and Steel Slag

As shown in Figure 4, after the expansion test of 20FASS, the surface of the expansive steel-slag particle was covered with a white material. According to an earlier study [41], these steel-slag particles contained a large amount of dead-burnt lime. Generally, f-CaO exists in the dead-burnt lime of steel slag. The XRD results for the white powder are shown in Figure 11. The main components of the white powder were found to be CaCO3 and Ca(OH)_2_. Most of the Ca(OH)_2_ was carbonized into CaCO_3_ due to long-term exposure to the air. Thus, the f-CaO enriched with steel slag was hydrated to form Ca(OH)_2_, which increased the volume of the solid phase. The increased volume generated an expansion stress between the steel-slag particles, leading to bulging.

Figure 12, Figure 13 and Figure 14 present the test results for the cement-fly-ash binder in Figure 4. Figure 13 shows the SEM test results for the binder on the side, where the binder contacted the steel slag. The results show that many acicular, fibrous, and plate-like C-S-H phases were formed on the surface of the binder. The EDS results of the three markers in the SEM image are shown in Figure 14. It can be seen that the EDS results of the three markers were in the C-S-H phase. No large crystals of Ca(OH)_2_ were formed by the hydration of f-CaO from steel slag [42] In addition, the XRD results in Figure 12 also show that calcium silicate and calcium aluminosilicate were mainly produced by hydration of the binder. The XRD pattern shows some peaks of amorphous phase, because the binder consisted of some fine powder of steel slag, and there was no characteristic peak of Ca(OH)_2_. This proved that the cement-fly-ash binder could react with the expansive components produced by the hydration of the steel slag [43,44].

In general, the mechanism by which fly ash inhibits the expansion of steel slag is due to the following aspects: the increase of fly ash content reduced the possibility of the formation of a skeleton by the steel slag, and prevented the concentration of expansion stress between the steel slag; and the low elastic modulus of the cement-fly-ash binder alleviated the expansion stress formed by the Ca(OH)_2_ expansion component, and also prevented its further growth by reacting with the Ca(OH)_2_. The specific process is shown in Figure 15.

## 4. Conclusions

The influence of different contents of fly ash on the mechanical properties and expansion behavior of cement-stabilized steel-slag base material were studied. In order to reveal the mechanism by which fly ash inhibits the expansion of steel slag, X-CT technology was used to explore the influence of macro structure on the expansion behavior of the stabilized material, and SEM-EDS and XRD were used to reveal the reaction principle between the fly ash and the steel slag. The major conclusions of the study are as follows:(1)The unconfined compressive strength, splitting strength, and compressive modulus of resilience of the cement-fly-ash-stabilized steel slag increased initially and subsequently decreased with an increase in fly-ash content. It reached a peak value when the content of fly ash was 20% (20FASS). In general, the cement-fly-ash-stabilized steel slag (30FASS) with 30% fly ash content not only exhibited a good volumetric stability, but also possessed good mechanical properties. Its compressive strength at 7 d, 28 d and 90 d was 4.31 MPa, 8.57 MPa, and 12.5 MPa, respectively.(2)The average expansion rate, dispersion rate, and number of bulges of the cement-fly-ash-stabilized steel slag decreased with an increase in the concentration of the fly-ash content. The volumetric stability of the stabilized material increased gradually. The cement-fly-ash-stabilized steel slag (fly-ash content ≥30%, suspension type) exhibited a low expansion rate, and the stabilized material was not damaged during the expansion process.(3)The mechanism by which the fly ash inhibited the expansion of the steel slag had two aspects. (A) The increase of fly-ash content possibly reduced the probability of the formation of a skeleton in the steel-slag particles, and the binder wrapped with steel slag produced an elastic-plastic deformation to accommodate the expansive component of the steel slag to alleviate the expansion stress. (B) Fly ash reacted with the expansive component of the steel slag.(4)Currently, relevant research is focused on cement-stabilized steel slag, which can be easily damaged due to expansion. Our study revealed the inhibition mechanism of fly ash on the expansion behavior of the cement-stabilized steel slag, which provides a theory and reference for the use of steel slag. However, there are many kinds of steel slag and fly ash, and the reaction rate of different types of fly ash and steel slag can vary. Future research needs to be focused on the inhibition effect of different kinds of fly ash on the expansion of different kinds of steel slag. Also, the raw material standard of cement-fly-ash-stabilized steel slag should be established to increase the application of steel slag in road-base materials.

## Figures and Tables

**Figure 1 materials-14-00495-f001:**
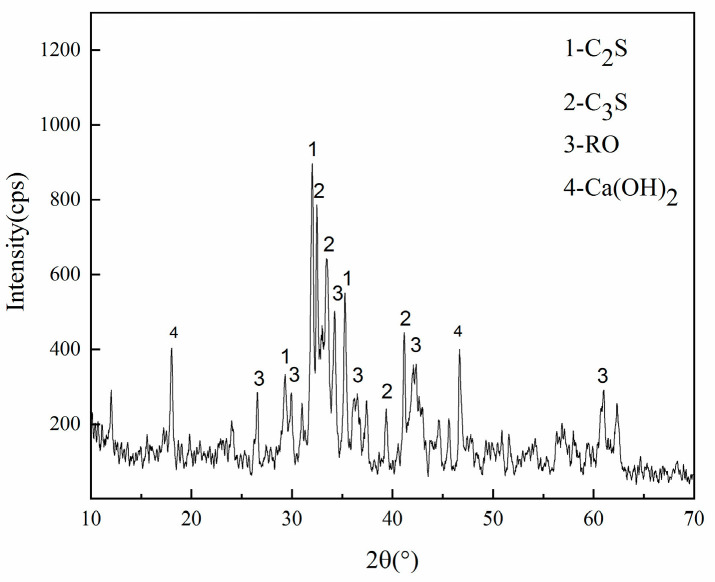
The XRD patterns of the steel slag.

**Figure 2 materials-14-00495-f002:**
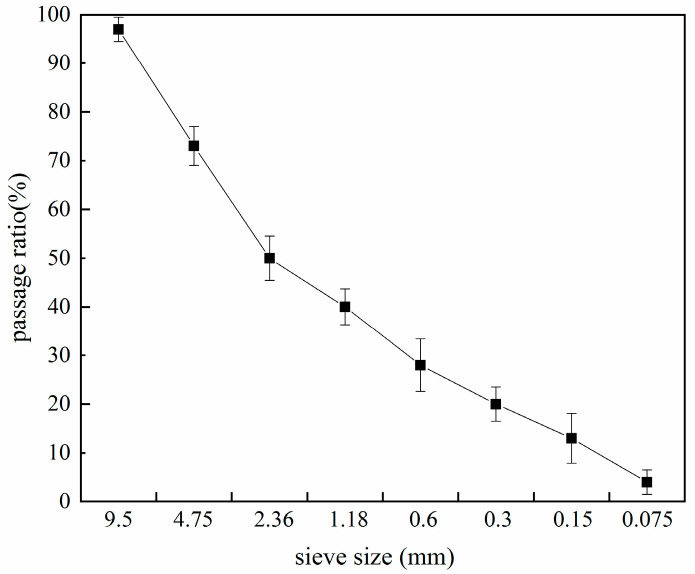
Particle-size distribution of the steel slag.

**Figure 3 materials-14-00495-f003:**
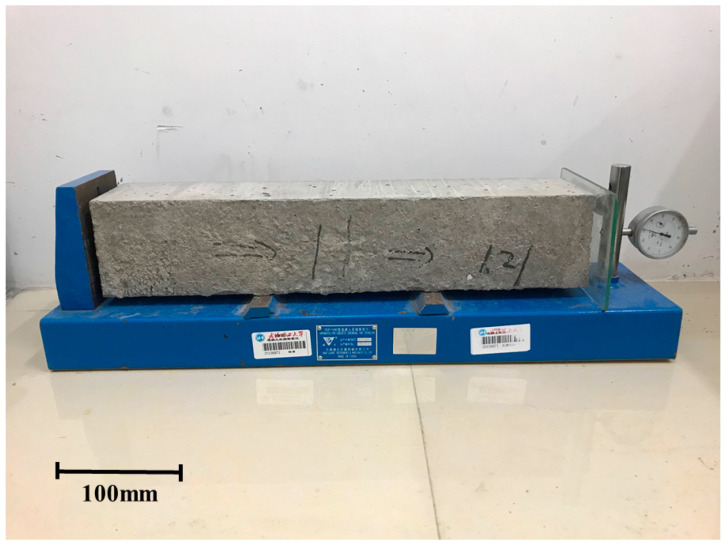
The expansion test.

**Figure 4 materials-14-00495-f004:**
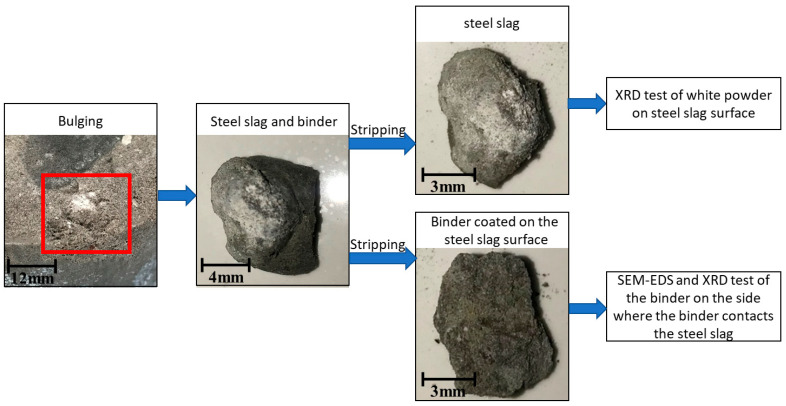
The sample handling and testing process.

**Figure 5 materials-14-00495-f005:**
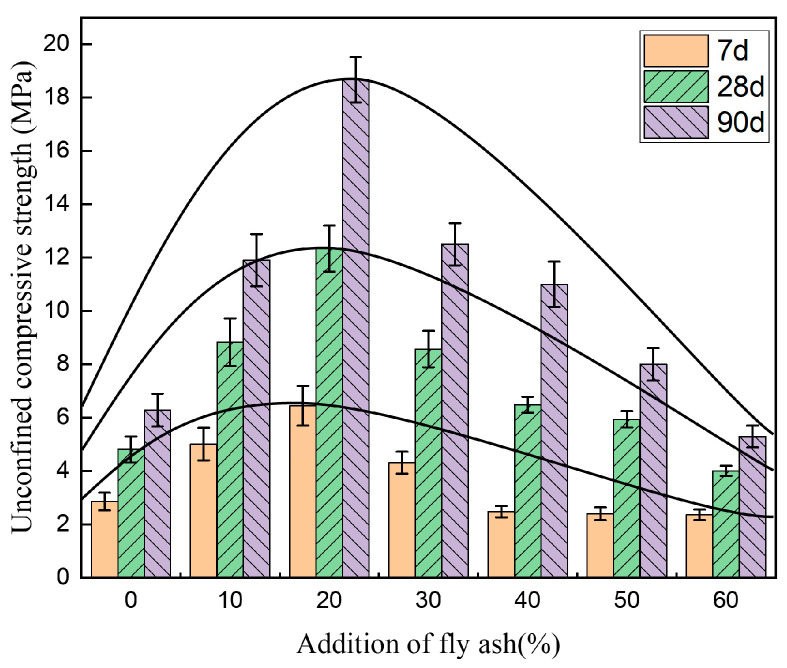
The effect of fly-ash addition on the unconfined compressive strength of the cement-stabilized steel slag. The error bars represent the standard deviation of the test results for nine samples.

**Figure 6 materials-14-00495-f006:**
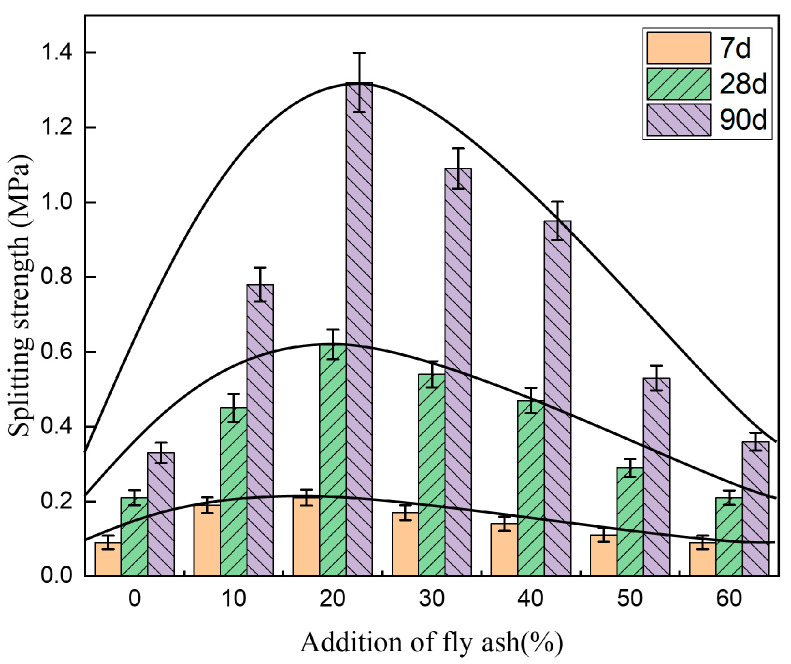
The effect of fly-ash addition on the splitting strength of the cement-stabilized steel slag. The error bars represent the standard deviation of the test results for nine samples.

**Figure 7 materials-14-00495-f007:**
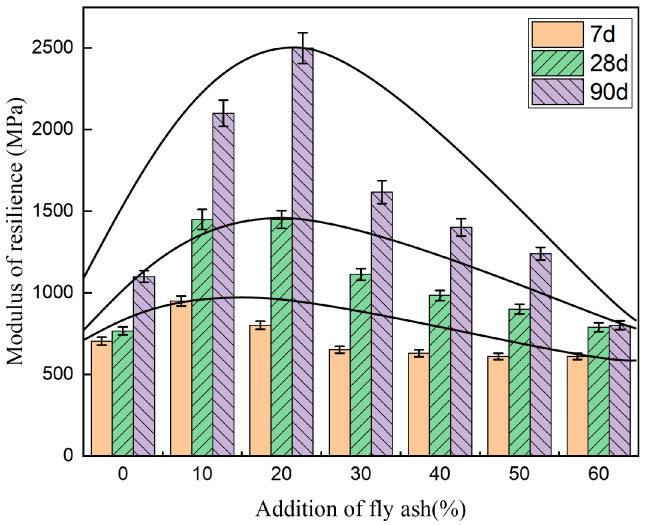
The effect of fly-ash addition on the compressive modulus of resilience of the cement-stabilized steel slag. The error bars represent the standard deviation of the test results for nine samples.

**Figure 8 materials-14-00495-f008:**
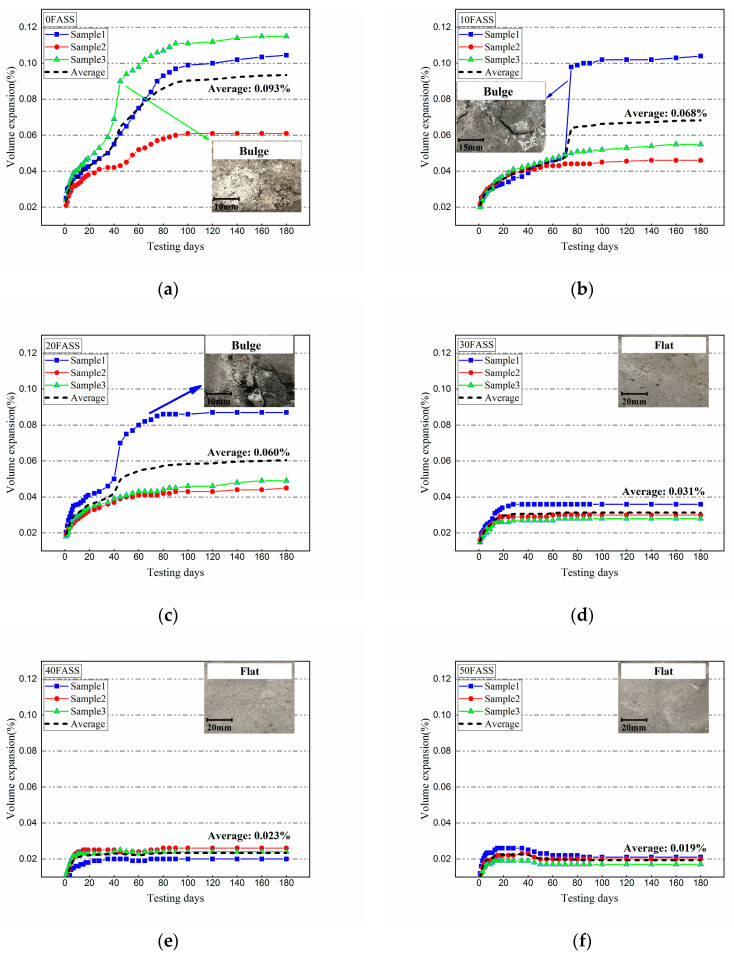
The expansion rates of 0FASS, 10FASS, 20FASS, 30FASS, 40FASS, 50FASS, 60FASS (**a**–**g**, respectively).

**Figure 9 materials-14-00495-f009:**
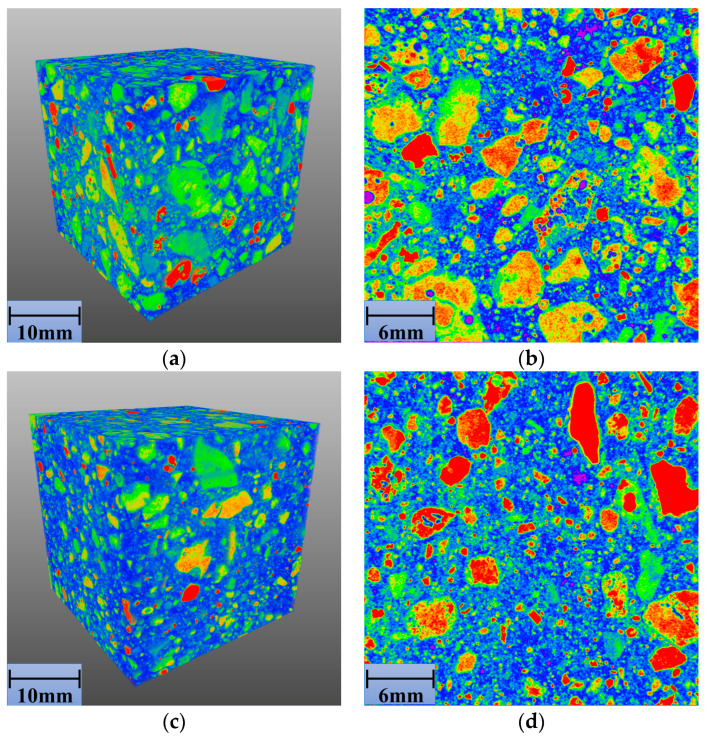
(**a**) 3D X-CT image of 0FASS, (**b**) 2D X-CT image of 0FASS, (**c**) 3D X-CT image of 20FASS, (**d**) 2D X-CT image of 20FASS, (**e**) 3D X-CT image of 40FASS, (**f**) 2D X-CT image of 40FASS.

**Figure 10 materials-14-00495-f010:**
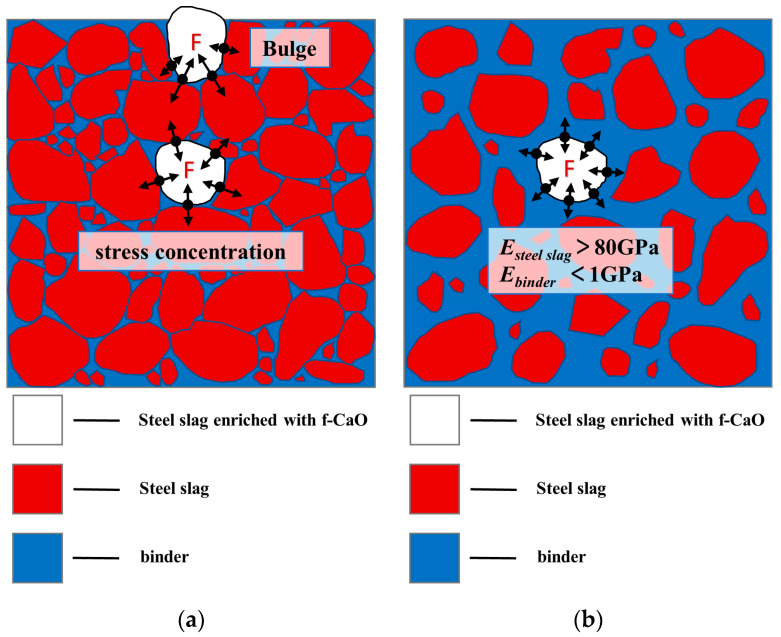
(**a**) Simulation diagram of the cement-fly-ash-stabilized steel-slag material (skeleton type) (**b**) Simulation diagram of the cement-fly-ash-stabilized steel-slag material (suspension type). *E_steel slag_* is the elastic modulus of the steel slag, *E_binder_* is the elastic modulus of the binder.

**Figure 11 materials-14-00495-f011:**
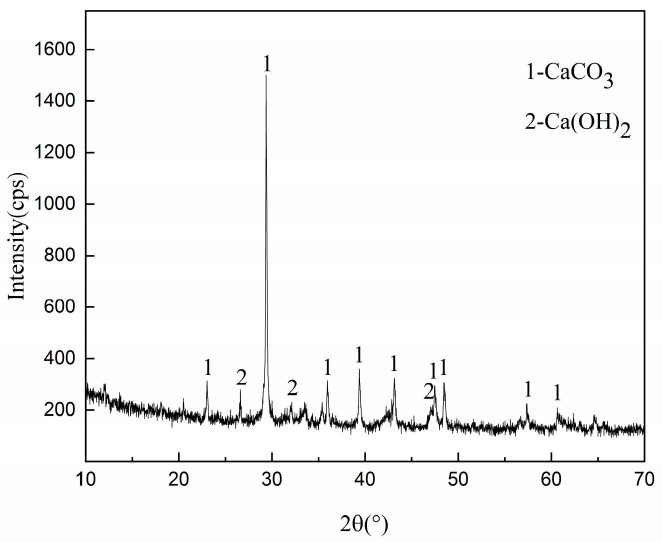
XRD Spectrum of the white powder.

**Figure 12 materials-14-00495-f012:**
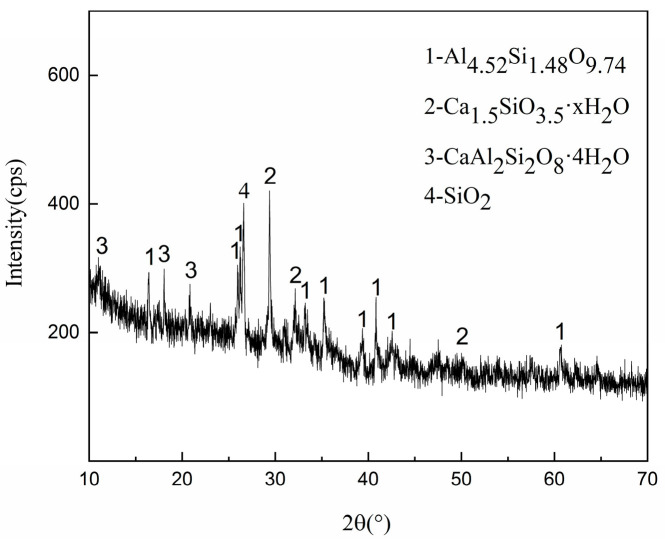
The XRD patterns of the binder in contact with the steel slag.

**Figure 13 materials-14-00495-f013:**
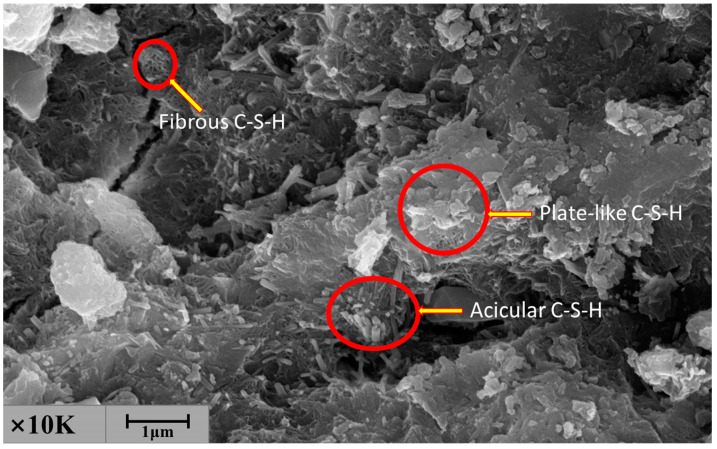
The SEM analysis of the binder on the side where the binder contacted and the steel slag.

**Figure 14 materials-14-00495-f014:**
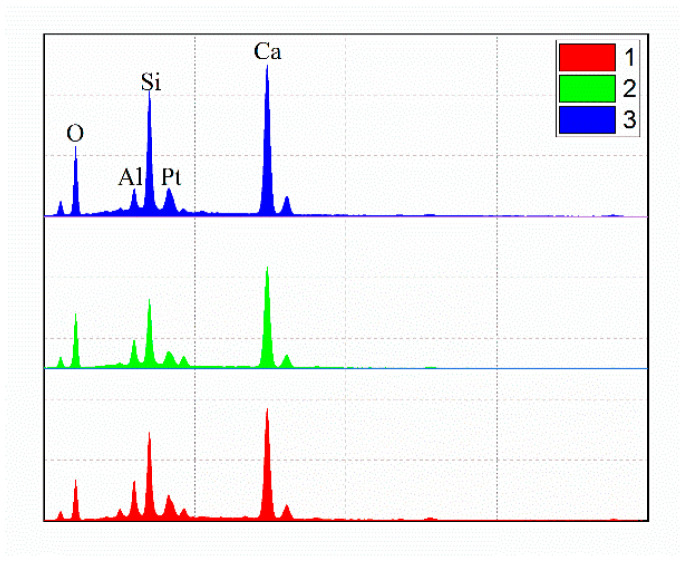
The EDS results for three marks in the SEM image.

**Figure 15 materials-14-00495-f015:**
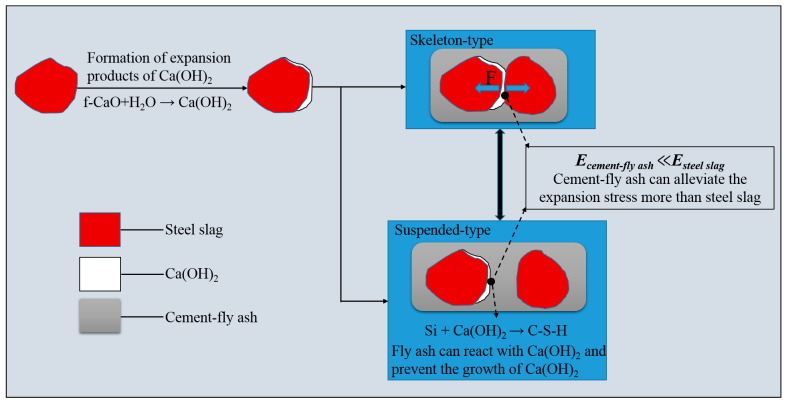
Schematics of fly ash inhibiting the expansion of the steel slag.

**Table 1 materials-14-00495-t001:** The chemical compositions of steel slag, fly ash, and cement (%) (error ± 0.05%).

Materials	LOSS	Fe_2_O_3_	Al_2_O_3_	CaO	SiO_2_	MgO	TiO_2_	K_2_O	SO_3_	P_2_O_5_
Steel slag	0.72	25.02	6.07	42.72	15.77	4.79	0.67	0.08	0.50	0.86
Fly ash	2.97	6.13	31.71	3.45	50.94	0.61	1.14	1.49	0.54	0.19
Cement	3.418	3.34	6.65	57.92	21.96	2.57	0.30	0.40	2.46	0.17

**Table 2 materials-14-00495-t002:** The basic properties of the steel slag.

Property	Steel Slag
Apparent density (cm^3^)	3.14–3.28
Water absorption rate (%)	4.44–4.52
Void rate (%)	36.3–38.1
f-CaO (%)	0.71–86
Water-immersion expansion rate (%)	0.06–1.25

**Table 3 materials-14-00495-t003:** The mechanical properties of the cement.

Setting Time (min)		Flexural Strength (MPa)		Compressive Strength (MPa)		Soundness
Initial setting	Final setting	3d	28d	3d	28d	Qualified
165	260	5.5	12.5	24.4	48.5	

**Table 4 materials-14-00495-t004:** Mix proportion, structure type, maximum dry density, and optimum moisture content of the cement-fly-ash-stabilized steel slag.

Number	Cement: Fly Ash: Steel Slag Ratio	Filling Factor	Structure Type	Maximum Dry Density (kg·m^3^)	Optimum Moisture Content (%)
0FASS	4:0:100	0.13	Skeleton-pore	2.30	7
10FASS	4:10:90	0.50	Skeleton-pore	2.29	7.609
20FASS	4:20:80	0.96	Skeleton-dense	2.26	9.5
30FASS	4:30:70	1.56	Suspended-dense	2.11	10.01
40FASS	4:40:60	2.33	Suspended-dense	2.03	12.09
50FASS	4:50:50	3.52	Suspended-dense	1.78	12.6
60FASS	4:60:40	5.26	Suspended-dense	1.64	15.8

**Table 5 materials-14-00495-t005:** The expansion rate, average value of expansion rate, coefficient of variation, and number and description of bulges for the samples on the 180th day of the test.

Addition of Fly Ash (%)	Sample No.	Expansion Rate (%)	Average (%)	Dispersion Coefficient (C_V_)	Description	Number of Bulges
0%	1	0.104	0.093	0.250	Bulge/crack	6
	2	0.086			bulge	
	3	0.115			bulge	
10%	1	0.104	0.068	0.373	bulge	3
	2	0.046				
	3	0.055			bulge/crack	
20%	1	0.087	0.060	0.314	bulge	1
	2	0.045				
	3	0.049				
30%	1	0.032	0.031	0.109	flat	0
	2	0.03				
	3	0.028				
40%	1	0.02	0.023	0.107		0
	2	0.026			flat	
	3	0.024				
50%	1	0.021	0.019	0.090		0
	2	0.02			flat	
	3	0.015				
60%	1	0.018	0.018	0.045		0
	2	0.019			flat	
	3	0.017				

## Data Availability

The data used to support the findings of this study can be made available from the corresponding author upon request.

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
