# Peer review of "Studies on the Volumetric Stability and Mechanical Properties of Cement-Fly-Ash-Stabilized Steel Slag"

_materials, 2021, doi:10.3390/ma14030495_

Round 1

Reviewer 1 Report

The manuscript entitled: Study on volumetric stability and mechanical properties of cement-fly ash-stabilized steel slag deals with the utilization of steel slag as a potential source of raw materials along with testing their volumetric stability and mechanical properties. I have the following concerns with the present manuscript:

  • How will the quality of the steel slag be ensured every time? If not how will be the quality of the final component is ensured?
  • The mechanism of fly ash inhibiting the expansion is steel slag is not convincing. It may be explained well with schematics.
  • The reaction mechanism of steel slag and fly ash is also not convincing. A schematics may be introduced for better understandability. 
  • A representative compressive stress-strain curve should be introduced in Figure 5.
  • Y-axis legend in Figures 1, 11, and 12 is missing
  • Error bars should be introduced for all data points in Figure 2, Tables 1, 2, 5
  • Figures 1, 2, and 8 are not of publishable quality.
  • All the microstructures in Figure 8 should be accompanied by scale bars.
  • Scale bar should be introduced in Figures 3, 4, 9 and 13.
  • All the typos in the manuscript should be rectified. For instance, space should be introduced between text/number and their units.
  • The English language needs attention.

Reviewer 2 Report

In my opinion, this is a very interesting work and it can be accepted for publication with minor changes. Only, I recommend that the authors modify the conclusions section, including an introductory paragraph to the conclusions they present, and finally a general conclusion to indicate the most relevant advance of the work and the future line of research to follow

Reviewer 3 Report

The authors presented interesting research. The structure of the manuscript appears to be well-organized: the introduction is supported by literature. The body of the paper is developed in all necessary parts in order to explain step by step the research philosophy and hence the obtained results. I have only a few minor substantive comments:

According to my knowledge, the unit of the modulus of resilience is not MPa (shown in Figure 7) but J/m3 (this is the strain energy per volume)

The content of Figure 9 is somewhat unclear. The drawing needs a better explanation in the text (meaning of colors).

There is no information about which samples are related to the XRD and SEM test results shown in Figures 12, 13 and 14.

Line 298 - Whether the white powder appearance was only for the 20FASS sample? If the cement-fly ash binder reacts with the products of the hydration of steel slag, the amount of white powder in the samples rich in fly ash should be smaller. There is no such comparison in the manuscript. The authors only performed qualitative, not quantitative, analysis.

The weakest point of the article is the quality of the English language. In my opinion manuscript requires extensive English correction.

Round 2

Reviewer 1 Report

The authors have partially addressed my comments. However, in the context of the present manuscript and the comments raised by other reviewers, I can recommend the manuscript for publication in the present form.